# Hormone-Independent Mouse Mammary Adenocarcinomas with Different Metastatic Potential Exhibit Different Metabolic Signatures

**DOI:** 10.3390/biom10091242

**Published:** 2020-08-27

**Authors:** Daniela Bispo, Victoria Fabris, Caroline A. Lamb, Claudia Lanari, Luisa A. Helguero, Ana M. Gil

**Affiliations:** 1Department of Chemistry and CICECO—Aveiro Institute of Materials (CICECO/UA), University of Aveiro, Campus Universitário de Santiago, 3810-193 Aveiro, Portugal; d.bispo@ua.pt; 2IByME—Instituto de Biología y Medicina Experimental, Vuelta de Obligado 2490, Buenos Aires C1428ADN, Argentina; ibyme@ibyme.org.ar (V.F.); carolinealamb@gmail.com (C.A.L.); lanari.claudia@gmail.com (C.L.); 3iBIMED—Institute of Biomedicine, Department of Medical Sciences, Universidade de Aveiro, Agra do Crasto, 3810-193 Aveiro, Portugal; luisa.helguero@ua.pt

**Keywords:** endocrine breast cancer, murine models, metabolism, NMR, metabolomics, hormone-independent growth, metastatic potential, medroxyprogesterone acetate

## Abstract

The metabolic characteristics of metastatic and non-metastatic breast carcinomas remain poorly studied. In this work, untargeted Nuclear Magnetic Resonance (NMR) metabolomics was used to compare two medroxyprogesterone acetate (MPA)-induced mammary carcinomas lines with different metastatic abilities. Different metabolic signatures distinguished the non-metastatic (59-2-HI) and the metastatic (C7-2-HI) lines, with glucose, amino acid metabolism, nucleotide metabolism and lipid metabolism as the major affected pathways. Non-metastatic tumours appeared to be characterised by: (a) reduced glycolysis and tricarboxylic acid cycle (TCA) activities, possibly resulting in slower NADH biosynthesis and reduced mitochondrial transport chain activity and ATP synthesis; (b) glutamate accumulation possibly related to reduced glutathione activity and reduced mTORC1 activity; and (c) a clear shift to lower phosphoscholine/glycerophosphocholine ratios and sphingomyelin levels. Within each tumour line, metabolic profiles also differed significantly between tumours (i.e., mice). Metastatic tumours exhibited marked inter-tumour changes in polar compounds, some suggesting different glycolytic capacities. Such tumours also showed larger intra-tumour variations in metabolites involved in nucleotide and cholesterol/fatty acid metabolism, in tandem with less changes in TCA and phospholipid metabolism, compared to non-metastatic tumours. This study shows the valuable contribution of untargeted NMR metabolomics to characterise tumour metabolism, thus opening enticing opportunities to find metabolic markers related to metastatic ability in endocrine breast cancer.

## 1. Introduction

Breast cancer (BC) is the leading cause of cancer deaths among women, accounting for an estimated 15% of all cancer deaths worldwide in 2018 [1]. BC classification and treatment are based on common histological features and expression of estrogen receptor alpha and progesterone receptors (ER and PR, respectively), as well as on human epidermal growth factor receptor 2 (HER2) and the proliferation marker Ki67. Molecular gene expression signatures, besides recognising major molecular subtypes, provide additional prognostic value, while giving insight into BC heterogeneity [2,3]. These approaches cannot, however, fully explain tumour metabolic characteristics [4]. Indeed, metabolic reprogramming may lead to significant heterogeneity between and within tumours [5]. This inter- and intra-tumour heterogeneity, usually illustrated by cellular, molecular or histopathological parameters [6], often correlates with prognosis, therapy response and survival rates [7,8]. Therefore, characterisation of tumour metabolism can provide valuable information about the tumour needs for survival, while disclosing therapeutic targets and complementing tumour classification. Metabolic traits may be identified through metabolomic strategies, that constitute the comprehensive analysis of endogenous metabolites (in biological complex mixtures such as biofluids, tissues or cells) and their response to perturbation (e.g., disease) [9]. Metabolomics has been used to evaluate inter-tumour (inter-individual) heterogeneity, either in an attempt to correlate metabolic profiles of human breast tumours with histological features or immunohistochemical markers [10,11,12,13], or to help distinguish intrinsic molecular subtypes in murine BC models [14,15,16]. The identification of possible metabolomic-based subtypes, in addition to the already established BC intrinsic subtypes, has indicated that metabolic information may add to the understanding of BC inter-tumour heterogeneity, beyond transcriptomic-based analyses [4]. Indeed, some studies have already suggested possible breast tumour subtyping schemes based on metabolic profiling [17,18].

Intra-tumour heterogeneity is determined by intrinsic (e.g., genetic/epigenetic events) and extrinsic factors (e.g., microenvironment, therapeutic intervention) and allows for phenotypic plasticity, which leads to distinct degrees of drug resistance and metastatic potential [19]. Understanding intra-tumour heterogeneity can lead to improvements in therapeutic interventions based on the overall profile of tumour cell subpopulations, rather than solely on predominant ones [6]. Intra-tumour metabolic heterogeneity has been relatively less studied than other defining characteristics, such as gene expression or mutational landscape, albeit it constituting the functional expression of those mutational traits in co-evolution with the tumour microenvironment [20]. Most metabolomic studies carried out in this context have been guided by in vivo imaging techniques and magnetic resonance spectroscopy (MRS) [21,22,23,24,25], and only a few high resolution ex vivo metabolomic studies have been reported. Metabolic tumour compartmentalisation has been investigated, either in vivo and/or ex vivo, in lymphoma (different lipid markers found to characterise necrotic and non-necrotic tumour regions [22]), liver cancer (different metabolic profiles associated with distinct intra-tumour immune statuses) [26], lung cancer (glucose levels differing between higher and less perfused tumour areas) [27], sarcoma (specific lipid/protein signatures correlated with different grading characteristics within the tumour) [24], kidney cancer (different metabolic patterns associated with tumour portions with distinct drug sensitivity) [28] and BC (differential phospholipid intra-tumoural distribution possibly related to different degrees of proliferation, hypoxia and inflammation) [23]. Besides the latter study, based on mass spectrometry (MS) imaging [23], intra-tumour metabolic heterogeneity in BC has been assessed by direct analysis of the tissue using high resolution magic angled spinning (HRMAS) nuclear magnetic resonance (NMR) [29,30]. Central and peripheral tumour samples showed variability in phosphocholine (PC) and phosphoethanolamine (PE) contents, whereas central specimens and core biopsies showed variability in adipate, arginine, fumarate, glutamate, PC and PE.

This NMR metabolomics work builds on the previously described NMR studies of BC tissue [29,30] and on MS studies of different BC types [12,31], by reporting an untargeted metabolomics study of tumour extracts, to unravel the underlying biochemical aspects of inter- and intra-tumour heterogeneity. In particular, proton ^1^H-NMR metabolomics was used to characterise polar and lipophilic extracts obtained from mouse mammary adenocarcinomas induced by medroxyprogesterone acetate (MPA), a model that closely resembles human hormone receptor-positive BC. The MPA model comprises several ductal adenocarcinoma tumour lines and, here, we have focused on two hormone-independent (HI) tumour lines that express ER and PR [32] but have different metastatic capacity [33]. The aim of this work was, therefore, to assess metabolic differences between non-metastatic 59-2-HI and metastatic C7-2-HI tumour lines (inter-line heterogeneity), as well as between tumours from each line (inter-tumour heterogeneity), and within different areas of each individual tumour (intra-tumour heterogeneity). In this work, we identified the main metabolites defining the different levels of heterogeneity, discussed the potential underlying differences in metabolic pathways and advanced possible markers of metastatic behaviour for the murine tumours under study. 

## 2. Materials and Methods 

### 2.1. Syngeneic Tumour Model and Procedures Carried Out in Mice

The MPA-induced mouse mammary tumour model consists of several ductal adenocarcinoma tumour lines obtained in BALB)/c (Bagg Albino) mice by continuous administration of MPA [33]. After several passages, these initially hormone-dependent tumours (HD) develop the ability to grow independently of exogenous hormone administration (and, eventually, develop resistance to endocrine therapy throughout serial transplantation). In this work, two tumour lines that grow without exogenous hormone supply (hormone-independent, HI) were used (59-2-HI and C7-2HI) (Figure 1a). These tumours are invasive carcinomas that express ER and PR. They are both inhibited by antiprogestin treatment, although only C7-2-HI gives rise to lymph node and lung metastasis. Thus, in terms of histology and endocrine growth, the MPA tumour model is one of a few murine models that closely resemble human hormone receptor-positive BC. Both 59-2-HI and C7-2-HI tumour lines were subcutaneously transplanted into the left inguinal flank of two-month-old BALB/c female mice (3 animals for each tumour line) using a trocar and the resulting tumours were allowed to grow up to 1 cm in their longest dimension. One tumour was implanted per animal. Mice were fed *ad libitum* and kept in 12 h light/dark cycles. Tumours were excised and immediately stored in −80 °C until analysis. All animal procedures were performed at the IByME Animal facility, having been approved by the local Institutional Animal Care and Use Committee (Approval no. 030/2016, dated 24 June 2016) and complying with regulatory standards of animal ethics. To study intra-tumour heterogeneity, each frozen tumour was placed on a Petri dish on dry ice and quickly cut into sections with a scalpel. The spatial location of the incisions depended on the shape of the tumours (Figure 1b). All sections were stored at −80 °C. The letters A–F were used to distinguish between different tumours and numbers 1–4 identify the four sections (or octants) randomly chosen for analysis of each tumour, in this work (Figure 1). 

### 2.2. Tumour Extracts

Frozen tumour octants (average 50 mg per sample) were ground to a fine powder by mechanical maceration in cooled liquid N_2_ using a pestle and mortar. Upon grinding, each sample was returned to an Eppendorf tube and extracted using methanol-chloroform-water, as described elsewhere [34]. Briefly, each octant sample was homogenised in 500 μL of a cold solution of methanol and milliQ water (4:1, vortexed for 1 min), 400 μL of cold chloroform and 200 μL of cold milliQ water (vortexed for 1 min). After resting for 10 min at 4 °C, samples were centrifuged (5 min, 8000 rpm), phases were separated, lipophilic and hydrophilic (aqueous) extracts were dried under a nitrogen flow and under vacuum, respectively, and stored at −80 °C until analysis.

### 2.3. NMR Spectroscopy

Before NMR spectra acquisition, aqueous extracts were re-suspended in 650 µL phosphate buffered saline (100 mM phosphate, pH 7.4) previously prepared in D_2_O (99.9% deuterium) with 60 mM Na_2_HPO_4_, 40 mM NaH_2_PO_4_ and 0.1 mM 3-(trimethylsilyl)-propionic-2,2,3,3-d4 acid (TSP), for chemical shift referencing. Lipophilic extracts were re-suspended in 650 µL deuterated chloroform (99.8% deuterium) with 0.03% tetramethylsilane (TMS), again for chemical shift referencing. After vortex homogenisation, 550 μL of the solution was transferred to the NMR tube. NMR spectra were recorded on a Bruker Avance DRX 500 spectrometer, Rheistetten, Germany (at 298 K. Standard 1D spectra were acquired with the *noesypr1d* pulse sequence for aqueous samples, and with the *zg* pulse sequence for lipophilic samples, using 7002.801 Hz spectral width, 32 k data points, a 2.3 s acquisition time, a 2 s relaxation delay (d1), 100 ms mixing time (d8) and 512 scans. Each FID (free induction decay) was zero-filled to 32 k points, multiplied by a 0.3 Hz exponential line-broadening function prior to Fourier transform. Spectra were manually phased, baseline corrected, and chemical shifts referenced internally to TSP (aqueous extracts) or TMS (lipophilic extracts) at *δ* = 0.00 ppm. Peak assignments were based on literature [35,36,37] and spectral databases, such as the Bruker Biorefcode AMIX database and the human metabolome database (HMDB [38]) and Chenomx NMR Suite (Chenomx Inc, Edmonton, Canada). 

### 2.4. Statistics Analysis and Other Spectral Analysis

Multivariate analysis was applied to the full resolution ^1^H-NMR spectra, using SIMCA-P 11.5 (Umetrics, Umeå, Sweden), and excluding specific regions: (a) water (5.09–4.68 ppm) and TSP (0.13–0.00 ppm) for aqueous samples; (b) chloroform (7.50–6.96 ppm), residual methanol (3.57–3.36 ppm) and TMS (0.15–0.00 ppm) for lipophilic samples. Spectra were aligned using a recursive segment-wise peak alignment [39], to minimise chemical shift variations, and data were normalised to the total spectral area, which accounts for sample concentration differences. Principal component analysis (PCA, unsupervised analysis used to detect intrinsic clusters and outliers within the data set) and partial-least-squares discriminant analysis (PLS-DA, supervised analysis to maximise class discrimination) were performed after unit variance (UV) scaling of the spectra, which gives comparable weight to each data value [40]. The corresponding loading weights were obtained by multiplying each variable by its standard deviation and were coloured according to each variable importance to the projection (VIP) using Matlab R2012a. Relevant peaks were integrated from the original spectra using Amix 3.9.5 (Bruker BioSpin, Rheinstetten, Germany) and normalised to the total spectral area. The individual metabolites that most contributed to class separation were selected based on their statistical significance (*p* < 0.05 in the Wilcoxon rank-sum nonparametric test [41]) and effect size [42] (|ES| > 0.5 and ES error < 75%). For multiple testing, *p* values were adjusted using the Bonferroni correction [43]. Statistical tests and heatmaps were carried out using Python 3.6.5. Statistical total correlation spectroscopy (STOCSY) [44] was carried out in selected cases, to aid assignment of some peaks. In the analysis of NMR spectra of lipophilic extracts, calculation of average fatty acid chain length, unsaturation and polyunsaturation degrees was carried out as described in previous reports [45].

## 3. Results

### 3.1. ^1^H-NMR Spectra of Polar and Lipophilic Extracts of Breast Tumours

A representative spectrum of aqueous extracts from a murine MPA-induced HI mammary tumour (Appendix A) shows the predominance of lactate, creatine, choline compounds, *s*-inositol, alanine and taurine, along with many other less abundant metabolites, including in the low-field region (*ca. δ* 5.5–9), namely amino acids (e.g., histidine, phenylalanine, tyrosine), nitrogenous bases (e.g., uracil, hypoxanthine), nucleosides (adenosine/inosine and uridine) and organic acids (formate and fumarate). Overall, 40 polar compounds were identified in tumour polar extracts (Appendix A). These results were consistent with previous reports performed on human breast tumours by HRMAS NMR and analysis of perchloric acid extracts [46], in addition to reports on cell line extracts [47,48]. The spectra of lipophilic extracts (Appendix A) unveiled several different compound families (Appendix A), including specific fatty acids (FA) (namely, oleic, linoleic, arachidonic and docosahexaenoic acids), in addition to phosphatidylethanolamine (PtdEtn), phosphatidylcholine (PTC), sphingomyelin, triglycerides (TGs), 7-lathosterol (0.55) and plasmalogen moieties (*δ* 5.90). To our knowledge, this is the first detailed ^1^H-NMR report of mammary tumour lipophilic extracts, building on previous studies, namely in vivo MRS of breast adipose tissue in BC patients [49], ex vivo ^31^P-NMR of mammary tumour extracts [50,51], ^1^H-NMR of cell extracts [35] and HRMAS NMR of tumours [52]. In the same context, other studies of lipophilic extracts have been carried out by MS for cell lines, animal models and human samples [53,54,55,56].

### 3.2. Metabolic Differences between 59-2-HI and C7-2-HI Tumour Lines

Unsupervised analysis by PCA (Figure 2a, left) indicated a clear difference in the metabolic profile of polar extracts between 59-2-HI and C7-2-HI tumours, and the corresponding PLS-DA model was significantly robust (*Q*^2^_cum_ = 0.892) (Figure 2a, right). PLS-DA loadings plots (Figure 2b) revealed a large number of metabolite differences between the two tumour lines and, upon integration and effect size calculation, statistically significant changes were noted for 31 identified metabolites (and many still unassigned resonances), mostly associated with *p*-values in the 10^−4^–10^−5^ range (Table 1). On the other hand, multivariate analysis of the ^1^H-NMR spectra of lipophilic extracts (Appendix A) revealed weak differences between tumour lines (*Q*^2^ < 0.5 for PLS-DA), although marked differences were observed for sphingomyelin and an unassigned resonance at *δ* 8.50 (Table 1).

Overall, the results show that the levels of 13 amino acids were significantly lower in 59-2-HI tumours, compared to C7-2-HI tumours, and glutamate was increased (Table 1). In addition, 59-2-HI tumours showed elevated levels of pyruvate and tricarboxylic acid (TCA) cycle intermediates (succinate, citrate and fumarate). Phospholipid metabolism exhibited distinct characteristics too, with elevated levels of choline, glycerophosphocholine (GPC) (although only with *p*-value < 0.05), glycerol and ethanolamine in 59-2-HI tumours, whereas phosphocholine (PC) levels were reduced. Other metabolite variations related to lipid metabolism included sphingomyelin and the ketone body acetone, with higher and lower levels, respectively, in 59-2-HI tumours. Furthermore, no changes in the average length or degree of unsaturation/polyunsaturation of FAs were noted between tumour lines (not shown). Nitrogen bases also showed a different pattern between tumour lines, namely including higher uracil levels in 59-2-HI tumours and a tendency for lower uridine (*p*-value < 0.05), as well as higher levels of inosine (consistently with elevated hypoxanthine levels) and, possibly, adenosine (singlet signal at *δ* 8.35 arises from either adenosine or inosine, Table 1). 59-2-HI tumours exhibited elevated levels of creatine and relative lower levels of *s-* and *m*-inositols. 

The heatmaps of the normalised integral values (Figure 3a,b) illustrate the differences in metabolite levels, which distinguish the two tumour lines, while also reflecting some inter- and intra-tumour differences for the metabolites shown (note that each row corresponds to a tumour section). The colour scale highlights most of the changes, whereas for some metabolites (e.g., methionine, fumarate, hypoxanthine) referral to the values in Table 1 is required, as the colour scale is not particularly illustrative. The overlaid spectra in Figure 3c exemplifies how clear some spectral ranges are between the two tumour lines, namely showing average lower PC/GPC ratios for 59-2-HI tumours, compared to C7-2-HI tumours. 

### 3.3. Metabolic Differences among Tumours of The Same Line

In order to investigate the significant metabolite changes between tumours of the same line, PCA and PLS-DA were carried out systematically for each tumour set (each set comprising four tumour sections), compared to the remaining tumour samples. Although all PLS-DA models were weak (*Q^2^* < 0.3), indicative of smaller variations relative to those identified between different lines (but also perhaps limited by the smaller group sizes), metabolites varying with statistical relevance (*p*-value < 0.05) in at least one of the models were identified and integrated for all samples (Appendix A). Interestingly, in 59-2-HI tumours, both polar and lipidic extracts showed significant variations, whereas in C7-2-HI tumours spectral changes were only clear for polar extracts. 

For 59-2-HI tumours, tumour A was distinguished from tumours B and C by higher levels of alanine (*p*-value 6.6 × 10^−3^) and proline (*p*-value 0.042) (and of a set of 4 unassigned aliphatic resonances, *p*-values 0.010–0.042); the profiles of tumours B and C were very similar, except for lower levels of ethanolamine in B, compared to the remaining tumours (Appendix A). Tumour A was also distinguished by relatively lower values of oleic, linoleic and arachidonic acids, as well as triglycerides; indeed, a lower average unsaturation degree was found for tumour A, along with a qualitative tendency for shorter chain lengths. C7-2-HI tumours also showed changes in alanine and proline (relative lower levels for tumour F, similarly to tumour A in the 59-2-HI family), this time accompanied by higher levels of aspartate, glutamate and glutamine, lower levels of lactate and PC, and higher levels of glucose. A set of unassigned resonances also contributed to distinguishing tumours E and F, from D, with a tendency for higher intensities of aliphatic resonances at *δ* 1.29, 2.20 and 2.68. Although no significant changes were picked up in lipophilic compounds for C7-2-HI tumours, tumour F showed a higher average polyunsaturation FA degree. 

### 3.4. Intra-Tumoural Metabolic Differences

Intra-tumour metabolic heterogeneity was evaluated mainly by careful and close inspection of the NMR spectra of tumour sections and, also by considering the metabolites identified in Appendix A as varying within each tumour (i.e., between lines corresponding to the different sections of each tumour). This enabled an overall list of varying identified metabolites (and unassigned resonances) for which the values of relative standard deviation (RSD) of intra-tumour variation could be calculated (Figure 4). Results indicated that, compared to C7-2-HI tumours, 59-2-HI tumours exhibited (i) higher intra-tumoural variation in asparagine, succinate, GPC and PC (and unassigned singlets at *δ* 1.30 and 2.89); and (ii) lower variability in histidine, uracil, hypoxanthine and adenosine/inosine, as well as in free cholesterol, oleic, linoleic and arachidonic acids. In addition, some metabolites were seen to vary significantly in most tumours, irrespective of the tumour line considered, namely formate, glucose, *s*-inositol, acetone, acetate and TGs.

## 4. Discussion

In this study, we used two mammary ductal adenocarcinoma tumour lines that express similar levels of ER and PR but have different metastatic potential [32,33]. Both 59-2-HI and C7-2-HI tumour lines are invasive ductal carcinomas, but only C7-2-HI spreads to axillary lymph nodes and lungs. This study aimed to investigate if different metabolic profiles can be associated to different metastatic potential and which metabolites are more prone to vary between and within tumours, for each line. As expected, profile differences were of larger magnitude between tumours of different lines, compared to inter-tumour variability within each line and intra-tumoural variability. The metabolite changes identified in the two tumour lines and their putative association to specific biochemical pathways are depicted in Figure 5. The important distinguishing features seem to impact mainly on glucose metabolism, nucleoside metabolism and lipid metabolism, possibly also involving antioxidant mechanisms. 

Regarding glucose metabolism, enhanced levels of pyruvate and TCA intermediates were noted in 59-2-HI tumours. Higher pyruvate levels are usually indicative of higher glycolytic rates; however, no changes were observed in lactate or glucose levels when compared to C7-2-HI. Thus, in non-metastatic 5959-2-HI tumours, the higher consumption of alanine may explain the enhanced pyruvate levels. Moreover, in -2-HI tumours, the clear depletion of anaplerotic amino acids, suggests their feeding into the TCA cycle. In addition, accumulation of intermediates citrate, fumarate and succinate may be an indication of slower TCA rates, as proposed previously in studies of the 67NR non-metastatic breast cancer cell line [57]. Citrate inhibits phosphofructokinase 1 (PFK1), pyruvate kinase (PK), pyruvate dehydrogenase (PDH) and succinate dehydrogenase (SDH). Therefore, the higher citrate levels in 59-2-HI tumours would inhibit pathways producing ATP [58] and thus reduce glycolysis. Consequently, compensation of pyruvate levels would occur through alanine consumption, followed by pyruvate accumulation because of PDH inhibition and enhancement of succinate and fumarate levels due to SDH inhibition [59,60]. Phe and Tyr metabolism could also contribute to fumarate levels (Figure 5). Furthermore, TCA retardation would result in slower NADH biosynthesis to reduce mitochondrial transport chain and ATP biosynthesis, the opposite observations being naturally applicable to metastatic C7-2-HI tumours. Notably, detached metastatic ovarian carcinoma cells overcome reduced glucose internalisation by elevating pyruvate intake, but also ATP synthesis and oxygen consumption [61], and it has recently been shown, at the transcriptomic level, that breast cancer micrometastases shift their metabolism to enhance mitochondrial oxidative phosphorylation during seeding [62].

Glutamate was the only detected amino acid observed to increase in 59-2-HI tumours. This can result from reduced glutamate dehydrogenase (GDH) activity and, since glutamine is not fed into the TCA cycle, such observation is consistent with slower TCA rate. GDH is activated by mTORC1 [63] and, given that in 59-2-HI there is a reduction of amino acids, it is likely that these tumours have lower mTORC1 pathway activation, compared to C7-2HI [64]. Reduced overall amino acids, including branched-chain amino acids (BCAA), and both essential and non-essential amino acids, may thus reflect anaplerotic consumption to feed the TCA cycle, as mentioned above. Moreover, the low levels of all three BCAAs may be due to a decrease in BCAA transaminase 1 (BCAT1) expression/activity, consistently, again, with lower mTORC1 activation in 59-2-HI tumours [65]. On the contrary, in C7-2-HI tumours, glutamate reduction may be related to glutamine feeding into the TCA cycle to promote oxidative phosphorylation and ATP biosynthesis, as well as with enhanced expression of glutamine synthetase (GS), to maintain a high glutamine pool for pyrimidine synthesis as observed in other tumours [66]. We hypothesise that this occurs in C7-2-HI tumours, consistently with the observed higher uridine levels. Indeed, luminal BC cells exhibit higher GS expression than basal-like BC cells, which supports the capacity of the tumour lines studied here (luminal-like) to induce GS expression [67]. In addition, C7-2-HI tumours may also import glutamine through solute carrier (SLC) transporters, some of which are positively regulated by MYC, whose signalling is enhanced in metastatic tumours [64]. 

In terms of choline compounds, the profile of the two lines was clearly distinct, with average lower PC/GPC ratios for 59-2-HI tumours, compared to C7-2-HI, indicating a clear shift in membrane phospholipids metabolism. Since 59-2-HI tumours have higher free choline levels and reduced PC levels, it is suggested that, in these tumours, choline kinase (CHK) activity may be reduced. However, enhanced GPC may possibly be related to higher phospholipase A2 (PLA2) and lyso-PLA1 in 59-2-HI, compared to C7-2-HI tumours, in an attempt to produce free fatty acids to compensate for lower TCA rate through β-oxidation. This would also explain the higher choline levels [68]. Other aspects related to lipid metabolism include higher levels of sphingomyelin in 59-2-HI tumours. Sphingomyelin is a precursor of ceramide, sphingosine-1-P and gangliosides. Deregulation in sphingomyelin levels can potentially impact on a variety of processes, including apoptosis, proliferation, inflammation and metastasis [69]. However, at this stage, it is not possible to advance any further details on the specific reasons for sphingomyelin changes. 

Interestingly, the two lines did not show any significant differences in global levels of fatty acids (FA), or FA average unsaturation degree and chain length, which probably means that any possible changes in the relative amounts of FA-containing families of compounds (expected when comparing metastatic with non-metastatic cells) may, in this case, not impact significantly on the nature of the composing FAs detected herein. Other metabolite changes are seen to distinguish 59-2-HI tumours from C7-2-HI, namely lower levels of *s-* and *m*-inositols (possible relation to local osmotic regulation [70] or synthesis of inositides with signalling roles in the cell [71]) and acetone, as well as higher creatine (energy metabolism) and hypoxanthine (antioxidant capacity) levels. 

As expected, differences between tumours from the same line and within tumour sections were of lesser magnitude, compared to inter-tumour differences between lines. A distinctive feature was the fact that 59-2-HI tumours differed importantly in oleic, linoleic and arachidonic acids and TG levels, as well as in average FA unsaturation degree and chain length; indeed, C7-2-HI tumours did not differ significantly in any detectable lipid moiety, only a weak variation in polyunsaturation degree having been noted. In 59-2-HI tumours, lower levels of FAs were accompanied by higher alanine and proline levels and changes in ethanolamine levels. Since energy requirements are variable within tumours because of differential proliferation, angiogenesis and microenvironment crosstalk, the lipid differences noted may reflect the higher dependence of 59-2-HI tumours on lipid catabolism to support lower TCA rates. Interestingly, C7-2-HI tumours exhibited a more variable polar composition, where tumours with lower levels of alanine and proline also showed higher levels of aspartate, glutamate, glutamine and glucose, and lower levels of PC and lactate, thus suggesting an important variation in glycolytic and fermentative rates between C7-2-HI tumours. 

The magnitude of intra-tumour metabolic heterogeneity was comparable between the two tumour lines, although partially based in distinct metabolites. Large intra-tumour variations common to the two lines comprised changes in formate, glucose, *s*-inositol, acetone, acetate and TGs (although no clear correlation was noted between these variations, they may reflect an interplay between glucose levels and the extension of ketone body synthesis). However, 59-2-HI tumours also showed distinct intra-tumoural metabolic patterns, compared to C7-2-HI, namely (1) larger variations in asparagine, succinate, GPC and PC (TCA cycle activity/anaplerosis and phospholipid metabolism) and (2) lower variations in histidine, uracil, hypoxanthine, adenosine/inosine (products of nucleotide degradation) and free cholesterol and FAs (lipid degradation). We suggest that metastatic HI tumours may be characterised by less changes in TCA and phospholipid metabolism and a different pattern comprising products of nucleotide and cholesterol and FA metabolism. 

## 5. Conclusions

This work presents, for the first time to our knowledge, an NMR metabolomics comparison of two mammary carcinoma tumour lines of different metastatic characteristics and, hence, prognostics. We have established a metabolic signature that strongly distinguishes the 59-2-HI (non-metastatic) and C7-2-HI (metastatic) lines, with basis on several features of glucose and amino acid metabolism, choline compounds profile, nucleotide metabolism and lipid metabolism. Putative interpretation of metabolomic results suggest that, compared to non-metastatic tumours, metastatic tumours seem to be characterised by relatively enhanced glycolysis and TCA activity, possibly resulting in faster NADH production and, hence, higher mitochondrial transport chain activity and ATP synthesis, along with a lesser need for lipids β-oxidation. Our results are also consistent with metastatic tumours showing glutamate depletion, all the above changes being consistent with enhanced mTORC1 activity. 

Within each tumour line, metabolic profiles differed significantly between tumours, the metastatic C7-2-HI tumours exhibiting more marked changes in polar compounds profile, suggesting differences in glycolytic capacity between tumours. Finally, intra-tumoural heterogeneity indicated that metastatic C7-2-HI tumours showed less changes in TCA and phospholipid metabolism and more marked variations related to nucleotide and cholesterol/FA metabolism, compared to non-metastatic tumours. It is clear that future studies are required in order to demonstrate/discard the putative hypotheses advanced here, however, this work demonstrates the valuable contribution of untargeted NMR metabolomics to help describe tumour metabolism at different levels, thus opening enticing opportunities to find metabolic markers of metastatic dissemination in endocrine BC. 

## Figures and Tables

**Figure 1 biomolecules-10-01242-f001:**
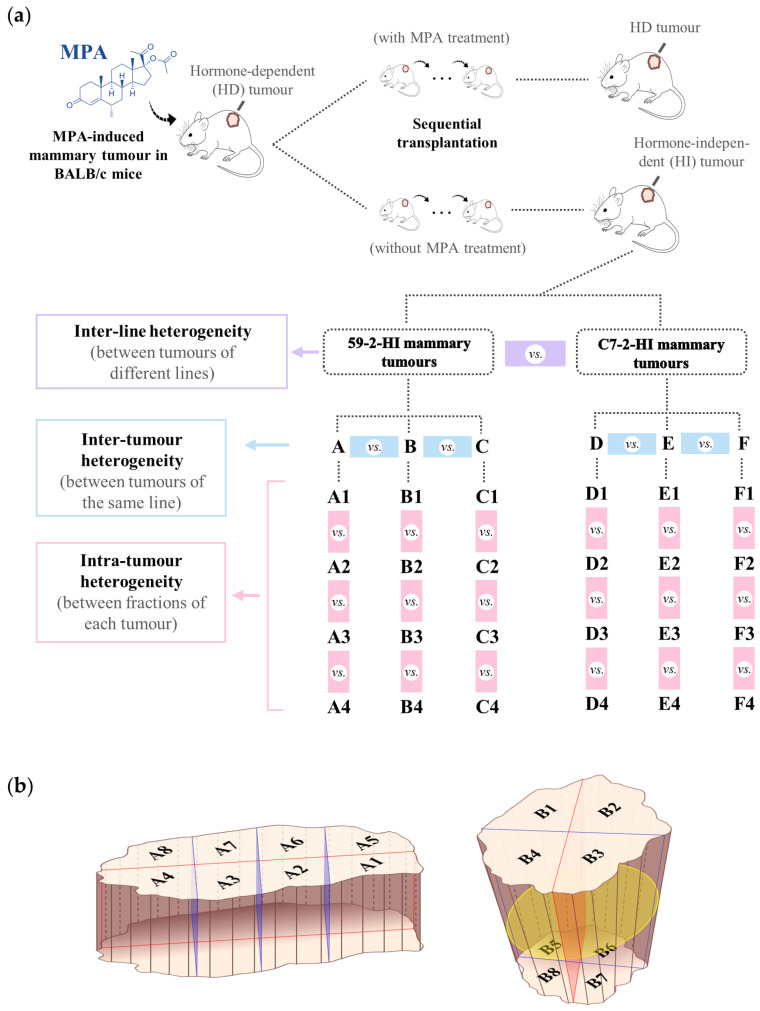
Brief description of the medroxyprogesterone acetate (MPA)-induced model of BC and mammary adenocarcinomas considered in this study. (**a**) Tumours were transplanted into syngeneic mice inoculated with an MPA depot to support hormone-dependent (HD) growth and into mice without MPA as control; some tumours began to grow without requiring MPA thus giving rise to hormone-independent (HI) lines: 59-2-HI (tumours A, B and C) and C7-2-HI (tumours D, E and F); (**b**) schematic representation of tumour sectioning criteria, depending on sample shape; every section included both peripheral and central parts; four sections of each tumour (1 to 4) were randomly chosen for NMR metabolomics.

**Figure 2 biomolecules-10-01242-f002:**
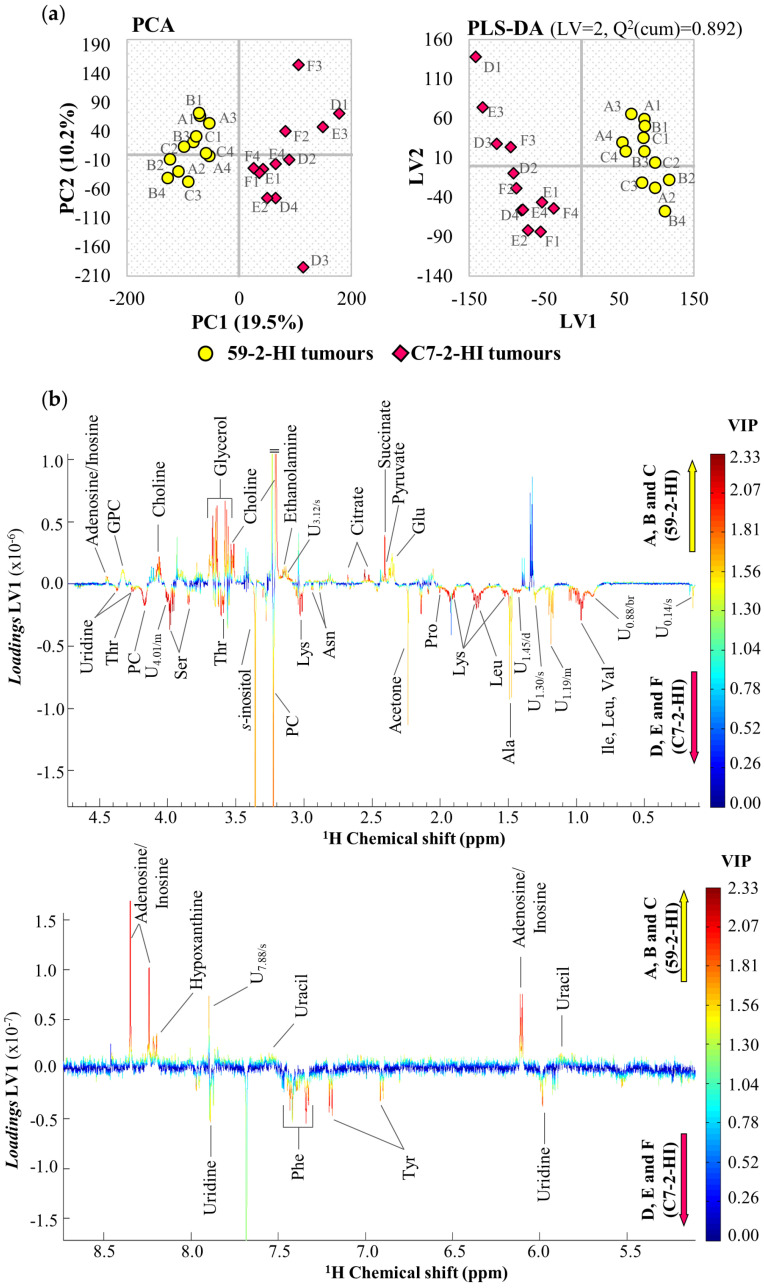
Results of multivariate analysis for comparison of 59-2-HI and C7-2-HI tumour lines. (**a**) Scores scatter plots for PCA and PLS-DA of 1H-NMR spectra of aqueous extracts from 59-2-HI and C7-2-HI tumour lines; and (**b**) LV1 loadings plot, coloured according to variable importance to the projection (VIP), corresponding to the PLS-DA model shown in (**a**). *Q^2^*(cum): cumulative predictive power; U*δ*/multiplicity: unassigned signal. 3-letter code used for amino acids; GPC: glycerophosphocholine, PC: phosphocholine.

**Figure 3 biomolecules-10-01242-f003:**
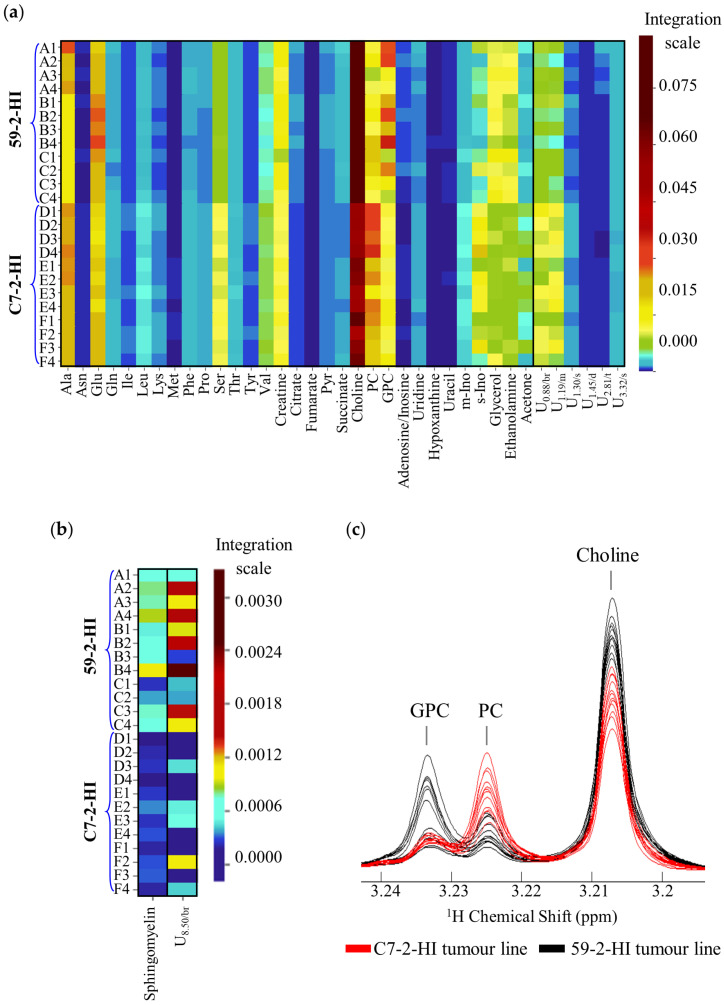
Heatmaps and example of spectral region representing statistically significant metabolic differences between 59-2-HI and C7-2-HI tumour lines. Heatmaps for (**a**) polar metabolites, and (**b**) lipophilic metabolites; the colour scale in the heatmaps varies from minimum (dark blue) to maximum (dark red) normalised integral values; (**c**) visual comparison of the ^1^H-NMR spectral region where choline compounds resonate. br, broad signal; d, doublet; dd, doublet of doublets; GPC, glycerophosphocholine; m, multiplet; PC, phosphocholine; s, singlet; t, triplet; U*_δ_*_/multiplicity_, unassigned signal.

**Figure 4 biomolecules-10-01242-f004:**
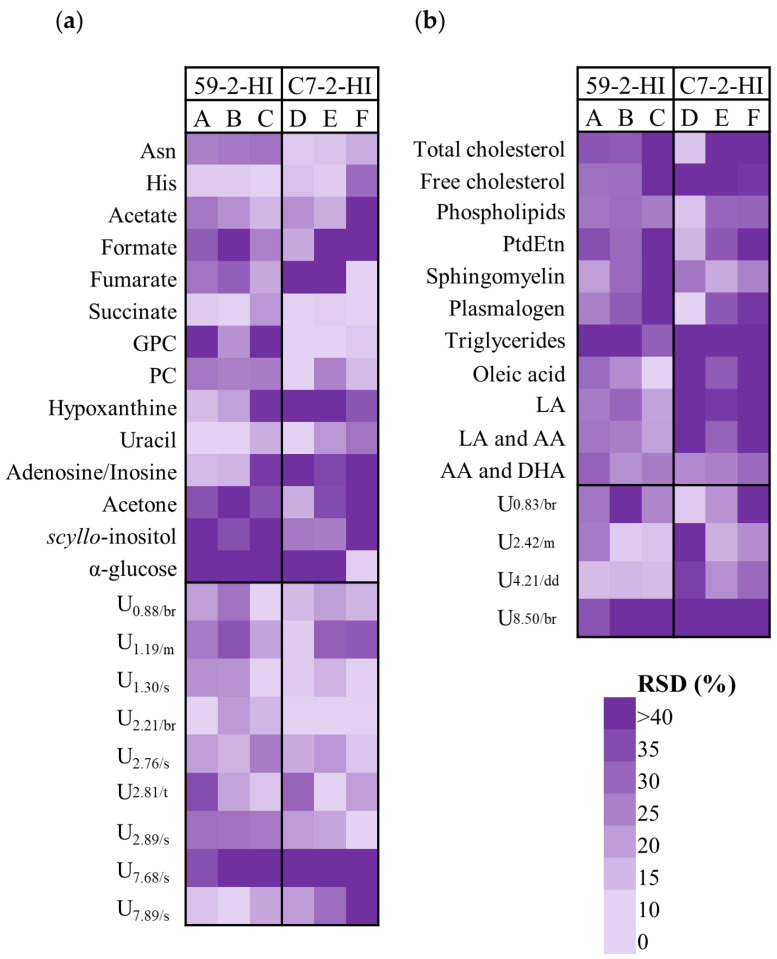
Heatmaps representing the % of relative standard deviation (RSD) of metabolites observed to vary most at the intra-tumour level. Only cases for which RSD > 20% for, at least, one of the tumours are illustrated, for (**a**) polar and (**b**) lipophilic extracts. 3-letter code used for amino acids; AA, arachidonic acid; DHA, docosahexaenoic acid; GPC, glycerophosphocholine; LA, linoleic acid; PC, phosphocholine; PtdEtn, phosphatidylethanolamine; RSD, relative standard deviation; U*_δ_*_/multiplicity_: unassigned signal.

**Figure 5 biomolecules-10-01242-f005:**
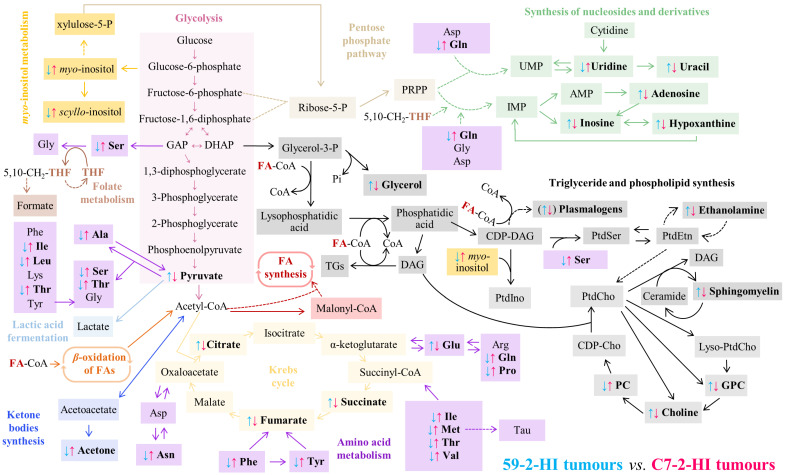
Schematic representation of the main metabolic differences found between 59-2-HI (blue arrows) and C7-2-HI (pink arrows) tumours. Metabolite names in bold correspond to those the levels of which were observed to change between the two lines (in the case of plasmalogens, the arrows in brackets indicate weak tendencies). AMP, adenosine monophosphate; CDP, cytidine diphosphate; CoA, coenzyme A; DAG, diacylglycerol; DHAP, dihydroxyacetone phosphate; FA, fatty acid; GAP, glyceraldehyde 3-phosphate; GPC, glycerophosphocholine; IMP, inosine monofosfato; PC, phosphocholine; PRPP, phosphoribosyl pyrophosphate; PtdCho, phosphatidylcholine; PtdEtn, phosphatidylethanolamine; PtdIno, phosphatidylinositol; PtdSer, phosphatidylserine; TGs, triglycerides; THF, tetrahydrofolate; UMP, uridine monophosphate.

**Table 1 biomolecules-10-01242-t001:** Statistically significant metabolic differences (|ES| > 0.50, ES error < 75% and *p*-value < 0.05) between 59-2-HI and C7-2-HI tumours. d: doublet; dd: doublet of doublets; m: multiplet; t: triplet; s: singlet; br: broad resonance; U*δ*/multiplicity: unassigned resonances, albeit contributing to the separation of the two tumour lines.

*δ*^1^H/ppm ^a^ (Multiplicity)	Metabolite	52-2-HI (A, B, C) vs. C7-2-HI (D, E, F)	
		Effect Size (ES) ^b^	*p*-Value ^c^
	**Polar compounds**		
1.48 (d)	Alanine	–1.78 ± 0.94	3.23 × 10^−3 d^
2.84 (dd)	Asparagine	–1.35 ± 0.89	1.11 × 10^−2 d^
2.36 (m)	Glutamate	1.67 ± 0.93	1.50 × 10^−3 d^
2.46 (m)	Glutamine	–1.54 ± 0.91	3.89 × 10^−3 d^
1.02 (d)	Isoleucine	–2.76 ± 1.12	8.64 × 10^−5^
0.97 (d)	Leucine	–4.12 ± 1.41	4.15 × 10^−5^
1.92 (m)	Lysine	–3.07 ± 1.18	3.23 × 10^−5^
2.65 (t)	Methionine	–1.57 ± 0.92	1.82 × 10^−3 d^
7.33 (m)	Phenylalanine	–3.37 ± 1.25	4.15 × 10^−5^
2.03 (m)	Proline	–2.77 ± 1.12	1.10 × 10^−4^
3.95 (dd)	Serine	–5.66 ± 1.79	3.23 × 10^−5^
4.25 (dd)	Threonine	–3.41 ± 1.25	3.23 × 10^−5^
6.91 (d)	Tyrosine	–2.04 ± 0.99	2.76 × 10^−4^
1.05 (d)	Valine	–2.92 ± 1.15	6.78 × 10^−5^
3.93 (s)	Creatine	1.18 ± 0.87	2.68 × 10^−3 d^
2.52 (d)	Citrate	6.59 ± 2.03	3.23 × 10^−5^
6.52(s)	Fumarate	1.64 ± 0.93	1.22 × 10^−3^
2.39 (s)	Pyruvate	1.78 ± 0.95	8.12 × 10^−4^
2.41 (s)	Succinate	3.18 ± 1.21	3.23 × 10^–5^
3.21 (s)	Choline	3.68 ± 1.31	3.23 × 10^−5^
3.22 (s)	Phosphocholine	–2.25 ± 1.02	2.20 × 10^−4^
2.23 (s)	Glycerophosphocholine	1.41 ± 0.89	2.82 × 10^−2 d^
8.35 (s)	Adenosine/Inosine	3.61 ± 1.30	3.23 × 10^−5^
7.86 (d)	Uridine	–1.27 ± 0.88	1.11 × 10^−2 d^
8.22 (s)	Hypoxanthine	1.98 ± 0.98	5.32 × 10^−4^
5.81 (d)	Uracil	2.38 ± 1.05	8.64 × 10^−5^
3.36 (s)	*scyllo*–inositol	–1.96 ± 0.97	4.29 × 10^−4^
3.63 (t)	*myo*–inositol ^e^	–3.21 ± 1.21	8.64 × 10^−5^
3.55 (dd)	Glycerol	2.48 ± 1.06	1.39 × 10^−4^
3.15 (t)	Ethanolamine	2.00 ± 0.98	2.20 × 10^−4^
2.24 (s)	Acetone	–1.90 ± 0.96	6.58 × 10^−4^
0.88 (br)	U_0.88/br_	–1.97 ± 0.98	4.29 × 10^−4^
1.19 (m)	U_1.19/m_	–1.85 ± 0.96	1.22 × 10^−3^
1.30 (s)	U_1.30/s_	–1.77 ± 0.94	4.29 × 10^−4^
1.45 (d)	U_1.45/d_	–3.06 ± 1.18	5.31 × 10^−5^
2.81 (t)	U_2.81/t_	1.28 ± 0.88	6.78 × 10^−5^
3.12 (s)	U_3.12/s_	1.95 ± 0.97	9.99 × 10^−4^
	**Lipophilic compounds**		
5.68 (m)	Sphingomyelin	2.54 ± 1.08	1.10 × 10^−4^
8.50 (br)	U_8.50/br_	1.51 ± 0.85	2.21 × 10^−3^

^a^ peak used for integration (part of the spin system); ^b^ Effect size (ES) calculated according to Berben et al. [42] (positive and negative ES values correspond to variations in 59-2-HI tumours, compared to C7-2-HI tumours); ^c^
*p*-values calculated according to the Wilcoxon Rank-sum test; ^d^
*p*-values that become no longer statistically significant upon Bonferroni correction [43], where cut-off *p*-values of 1.35 × 10^−3^ (aqueous extracts) and 2.5 × 10^−2^ (lipidic extracts) were used; ^e^ tentative assignment, as identification is hindered by signal overlap particularly in 59-2-HI tumours.

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
