# Peer review of "Hormone-Independent Mouse Mammary Adenocarcinomas with Different Metastatic Potential Exhibit Different Metabolic Signatures"

_biomolecules, 2020, doi:10.3390/biom10091242_

Round 1

Reviewer 1 Report

The paper presented by Gil and co-workers investigates metabolomic profiles of tumors derived from two cell lines, one metastatic and the other non-metastatic, using NMR-based untargeted metabolomics. The authors also describe their evaluation of different regions of the tumors (intra-tumor differences) and between tumors from different biological replicates.

The paper is well written but lacks statistical evaluation of two of its three aims. Although clear statistical differences were observed for the studied tumors, the approach used to describe both inter- and intra-tumor differences are questionable and should be deeply revised prior to publication.

The authors should also give information on the size of the tumors and the sample weight.

Here are the detailed topics:

  • In BC, intra-tumor metabolic heterogeneity has been assessed by direct analysis of the tissue using high-resolution magic angled spinning (HRMAS) Nuclear Magnetic Resonance (NMR) [29,30]. Authors should remember NMR is not the only technique for metabolomics. Mass Spectrometry has also been used for this purpose and no mention of that was done. Here it is one example, but authors are encouraged to find more: https://pubmed.ncbi.nlm.nih.gov/28193010/;
  • This sentence should be removed. First, its meaning is not clear. Else, mammary tumor lipophilic extracts have already been analyzed (https://www.ncbi.nlm.nih.gov/pmc/articles/PMC3245342/ , for example). “To our knowledge, this is the first detailed 1H NMR report of mammary tumour lipophilic extracts, building on in vivo MRS of breast adipose tissue of BC patients [48], ex vivo 31P NMR of mammary tumour extracts [49,50], 1H NMR of cell extracts [34] and HRMAS NMR of tumours [51].”
  • It is not clear how many mice were used per cell line. From Fig 1, I believe there were three per type, but it is not written in the text.
  • Why do the authors emphasize polar metabolites in lines 206-209 and on table 1 if they demonstrated that the most important differentiators are among the nonpolar metabolites? Else, nonpolar metabolites occupy only two lines on table 1? I would expect the opposite!
  • Why do the authors present non-significant metabolites in table 1, according to p-value? If they are not significant, should they be highlighted?
  • Figure 3 is not adding any information. The heatmap is not well illustrative. The authors should consider removing it.
  • How can the authors infer differences among tumors (item 3.3) if no technical replicates were used? How can they evaluate intertumoral %RSD if they didn’t evaluate the %RSD of any quality control sample, aimed at evaluating the reproducibility of their methods? The authors say they used visual inspection of the spectra to look for differences. It is not appropriate. PLS-DA is also not appropriate, since the tumors are identical, in theory.
  • Sup Fig 3 needs a table with the significance of the metabolites since the color scale is not quite representative.
  • 4 should come prior to 3.3
  • It is not clear why intra-tumoral heterogeneity was measured based on inter-tumoral heterogeneity results (Fig S3), in regards to the sentence: “Intra-tumour metabolic heterogeneity was evaluated by considering the metabolites identified in Figure S3 as well as by visual inspection of the spectra.”
  • Figure 4. It is not clear based on what samples the %RSD was calculated. For example, for tumor A, were the results from the eight octants used? And for Tumor F, four quadrants? That should be described.
  • The main questions were lost in the text. (1) to assess metabolic differences between non-metastatic 59-2-HI and metastatic C7-2-HI tumour lines; (2) differences between tumours from each line (inter-tumour heterogeneity), and (3) different areas of each individual tumour (intra-tumour heterogeneity). The statistical approach only enabled the authors to answer the first aim. The two others were not satisfactorily answered both due to experimental and statistical modeling approaches. It is not possible to conclude if central or peripheral tumor samples show different metabolomic profiles.

Reviewer 2 Report

The manuscript presents the NMR based metabolomics analysis of non-metastatic (59-2-HI) and metastatic (C7-2-HI) breast cancer lines and the comparison of their metabolome in order to discover biomarkers related to the metastatic ability. Intra-tumor heterogeneity The research is very well planned and very thoroughly executed. The results are clearly presented and the discussion covers all aspects. Overall the work is part of ongoing research of the group being an excellent example of NMR based metabolomic analysis and can be published as it is although this reviewer would suggest some minor clarifications.

In the methods section the number of samples analysed should be mentioned

The authors should mention in the conclusions phosphoscholine/glycerophosphocholine ratios as biomarker that was very interestingly shown in fig 3c

Although the NADH production has been underlined in the discussion and conclusions it is not clear if NMR could provide any relevant information. Signals at >8.5 ppm are present in the spectra however not assigned. The authors should clarify if these signals could be assigned to NAD+ or NADP.

This reviewer would strongly suggest to validate the mTOR related hypothesis, advanced by the authors, with western-blot analysis. Metabolomics should start correlate with proteins and transcriptomics in order to produce meaningful insights to the mechanisms.

In fig4 Hypoxantine should be corrected “hypoxanthine”

Reviewer 3 Report

Bispo and coauthors conducted a 1H-NMR-based metabolomics study to explore differences in the metabolic signatures of two hormone-independent mouse mammary adenocarcinomas with different metastatic potential. Although the topic of the study is certainly relevant, in the current form the study presents multiple issues that negatively affected the work and thus, in this form it can not be accepted for publication. Some of the major points that would need to be corrected/extended to make this study acceptable are reported below:

  • The methods section regarding the study design is not very well-described, however, from Figure 1 I understand that the study population consists of only 6 mice: 3 mice with 59-2-HI tumors and 3 mice with C7-2-HI tumors. It is evident that the sample size is underestimated, and this compromises all the work done. Although laboratory mice are genetically identical, the tumors are not identical, thus 3 mice per group are too few, and in order to make this study acceptable the study population must be increased (as an example, also for in-vitro study on cell-lines at least 5 independent replicates are requested).
  • Other important issue is the fact that authors consider the samples of the same tumor as independent samples, and this is wrong and affects the analysis by improving the classification accuracy. Samples of the same mouse in the same physiological conditions tend to cluster together, and so the accuracy could be enhanced by the individual recognition. In situations like this, the cross-validation (CV) should be properly adjusted removing in each step all the samples of the same mouse. Actually, it is not explained in the description of the statistical analysis if CV was performed, but Q2 is reported.
  • The paragraph related the intra-tumor variability is completely obscure both for results both for the methods utilized. How 1:1 sample can be compared? Why the authors decided to analyze only 4 randomly parts of the tumor?
  • Finally, I suggest authors to write section 2.4 clearly and with more details.

Round 2

Reviewer 1 Report

No coments